# Antioxidant Responses Induced by PFAS Exposure in Freshwater Fish in the Veneto Region

**DOI:** 10.3390/antiox11061115

**Published:** 2022-06-03

**Authors:** Elisabetta Piva, Sophia Schumann, Serena Dotteschini, Ginevra Brocca, Giuseppe Radaelli, Andrea Marion, Paola Irato, Daniela Bertotto, Gianfranco Santovito

**Affiliations:** 1Department of Biology, University of Padova, 35122 Padova, PD, Italy; elisabetta.piva.2@studenti.unipd.it (E.P.); sophia.schumann@studenti.unipd.it (S.S.); serena.dotteschini@studenti.unipd.it (S.D.); paola.irato@unipd.it (P.I.); 2Department of Comparative Biomedicine and Food Science, University of Padova, 35122 Padova, PD, Italy; ginevra.brocca@unipd.it (G.B.); giuseppe.radaelli@unipd.it (G.R.); 3Department of Industrial Engineering, University of Padova, 35122 Padova, PD, Italy; andrea.marion@unipd.it

**Keywords:** antioxidant enzymes, freshwater fish, oxidative stress, PFAS, Veneto Region

## Abstract

In recent decades, the interest in PFAS has grown exponentially around the world, due to the toxic effects induced by these chemical compounds in humans, as well as in other animals and plants. However, current knowledge related to the antistress responses that organisms can express when exposed to these substances is still insufficient and, therefore, requires further investigation. The present study focuses on antioxidant responses in *Squalius cephalus* and *Padogobius bonelli*, exposed to significant levels of PFAS in an area of the Veneto Region subjected to a recent relevant pollution case. These two ubiquitous freshwater species were sampled in three rivers characterised by different concentrations of PFAS. Several biomarkers of oxidative stress were evaluated, and the results suggest that PFAS chronic exposure induces some physiological responses in the target species, at both cellular and tissue scales. The risk of oxidative stress seems to be kept under control by the antioxidant system by means of gene activation at the mitochondrial level. Moreover, the histological analysis suggests an interesting protective mechanism against damage to the protein component based on lipid vacuolisation.

## 1. Introduction

In recent years, the scientific community is focusing on “emerging contaminants”, i.e., chemicals that are not commonly monitored in the environment but have the potential to enter it and cause known or suspected adverse ecological and human health effects. Within this group, the per- and polyfluorinated compounds (PFAS) stand out as major players.

PFAS are a heterogeneous group of highly fluorinated aliphatic substances, similar to the non-fluorinated analogues from which they derived, except that in their molecular structure fluorine atoms replace all hydrogen substituents. At least one “perfluoroalkyl moiety” –C_n_F_2n_– [1], therefore, characterises PFAS. Since the 1940–1950s, PFAS have been widely produced and manufactured all over the world [2]. Initially considered non-toxic and inert, these chemicals were subsequently associated with many adverse health effects, and concern about their toxicity increased exponentially.

Perfluorooctanoic acid (PFOA) and perfluorooctanesulfonic acid (PFOS) have been among the first to be detected in human blood and in the environment and are considered some of the most widespread PFAS, as shown in several studies [3].

Many of these synthetic chemical compounds are known to be both hydro- and lipophobic, as well as chemically and thermally stable [1]. They have also dielectric properties, low surface energy, low friction properties [3], and a surfactant nature [4]. PFAS are extremely resistant to hydrolysis, photolysis, biodegradation, and metabolism [5]. They are characterised also by high mobility and persistence, as they can travel great distances dispersing in surface and groundwater bodies, as well as in air in the form of dust particulates [6]. Their chemical and physical characteristics support their intensive use in many industrial and commercial applications such as firefighting foams, water and oil repellents, surfactant agents, lubricants, and coating agents [2].

Since the early 2000s, PFAS began to be detected in the environment, even far away from where they were originally used or manufactured. They have been traced as far as the Arctic and Antarctica [7]. PFAS have been detected in several different matrices—from aquatic ecosystems (surface and ground waters) to sediments, flora, and fauna [8]. In 2011, the Italian Ministry of Environment, Land, and Sea commissioned the control of PFAS contamination in the Italian water streams to the Institute of Water Research of the National Research Centre (IRSA-CNR) [9]. The results of these analyses were communicated in 2013, and it was found that in some provinces of the Veneto Region, drinking water samples exhibited extremely high PFAS concentrations. Twelve types of PFAS were identified, e.g., perfluorobutanoic acid (PFBA), perfluorobutanesulfonic acid (PFBS), and especially PFOA and PFOS [10].

Modern eco-physiological studies focus on if and how organisms can respond to environmental chemical stressors. The measurement of appropriate biomarkers is, therefore, crucial to determining the interaction between chemicals and biological systems. Toxicants, such as PFAS, act at the molecular level causing an alteration in reactive oxygen species (ROS) production, followed by a cellular redox imbalance. This can subsequently cause DNA damage, enzyme activity modifications, and a subsequent cascade of oxidative damage to the entire organism [11,12].

To protect cells from these damages, organisms have evolved various antioxidant multi-enzymatic systems [13,14,15,16,17]. Some of these enzymes can be used as good molecular biomarkers for contaminant-mediated oxidative stress and may also represent good markers of physiological response in organisms chronically exposed to xenobiotics [18].

The aim of the present research was first of all to identify the potential effects of chronic exposure to PFAS in freshwater fish and the induced physiological responses. A second goal was to verify if the extent of these effects correlates to different environmental PFAS levels. For this purpose, two ubiquitous species in the Veneto Region were studied: *Squalius cephalus* (European chub) and *Padogobius bonelli* (Padanian goby). These two species are particularly interesting because they occupy two different environmental niches and have different feeding habits. Therefore, they can be affected differently by PFAS and can express species-specific responses.

## 2. Materials and Methods

### 2.1. Sampling Sites 

The present research is based on a fieldwork activity that was carried out in March 2021. The three sampling sites were located in three different water streams in the province of Vicenza (Italy), characterised by different levels of PFAS contamination (monitoring data were provided by the Regional Agency for Environmental Prevention and Protection of Veneto, ARPAV, in 2020). 

The first location, referred to as Site 1, is Torrente Leogra in Torrebelvicino (45.71779193 N; 11.30627616 E). PFAS concentrations for this site were always below the detection threshold, so it was initially considered a control site. However, subsequent analyses aimed at verifying the presence of PFAS in fish residing in this environment have shown that even the control specimens have accumulated these contaminants in their tissues, albeit in very low concentrations [19].

The second site, referred to as Site 2, is Roggia Moneghina in Grumolo delle Abbadesse (45.51163342 N, 11.65604081 E). This site was characterised by a low level of PFAS contaminations, mostly represented by PFOS (95.5 ng/L). 

Finally, the third site is the Retrone river in Altavilla Vicentina (45.51350834 N, 11.5045197 E), characterised by a very high level of PFAS pollution (PFOA: 582.6 ng/L; PFOS: 95.5 ng/L). 

Given this evidence on pollutant levels, these three sites are referred to in this study as “Very low-polluted Site”, “Low-polluted Site”, and “Highly polluted Site”, respectively.

### 2.2. Samples Preparation

Ten specimens of each species of *S. cephalus* and *P. bonelli* were sampled from each of the three sites by electrofishing (authorised by decree of the director of the Agri-environment, Planning, and Management of Fish and Wildlife Hunting of the Veneto Region, n. 336 of 14 December 2020). For each species, the animals were of similar size: *S. cephalus* 19.99 cm (±5.14) and *P. bonelli* 6.81 cm (±0.65). After being caught, fish were immediately anesthetised and then euthanised with an overdose of essential clove oil, prepared in water-soluble form by dilution with ethyl alcohol (concentration 0.007 mL/L). All specimens were weighed and measured for total body length and dissected. The liver was removed, immediately frozen in liquid nitrogen, and stored at −80 °C until analysed. Liver samples of 6 individuals for each species and site were fixed also with 4% paraformaldehyde for histological analysis.

### 2.3. Primers Design, Total RNA Extraction, Sod2, and Gpx4 cDNA Synthesis

Amino acid and nucleotide sequences of *sod2* and *gpx4* from teleost fish obtained from the NCBI database were aligned by MUSCLE to identify conserved domains for primer design. At first, the species that are phylogenetically closer to *S. cephalus* and *P. bonelli* were identified. A similarity between *Periophthalmus magnuspinnatus* (mudskipper) and *P. bonelli* was identified, as these two species belong to the same family—namely, Gobiidae. In the same way, a similarity between *Pimephales promelas* (Fathead minnow) and *S. cephalus* was identified, since both belong to the same family—namely, Leuscidae.

The identities between the coding regions of the sequences were validated with a BLAST comparison (https://blast.ncbi.nlm.nih.gov/Blast.cgi (accessed on 1 March 2021)). The multiple alignments of the sequences with ClustalW2 led to the identification of the most conserved regions. Primers were designed in the coding regions, and the primer sequences were analysed with the IDT Oligo Analyser tool (https://eu.idtdna.com/pages/tools/oligoanalyzer (accessed on 1 March 2021)). Primer sets are shown in Appendix A.

Total RNA was purified from liver tissues of both species using PRImeZOL™ reagent (Canvax, Córdoba, Spain) according to the manufacturer’s protocol. Further purification was performed with 8M LiCl [20] to remove glucidic contaminants, and the quantification was performed using the NanoDrop ND-1000 spectrophotometer (Thermo Fisher Scientific, Waltham, MA, USA); RNA integrity was assessed by running an aliquot of RNA (1000 ng/μL) on a denaturing gel stained [21]. The cDNA synthesis was performed using a Biotechrabbit™ cDNA Synthesis Kit (Berlin, Germany) at 50 °C for 1 h + 99 °C for 5 min, from 1 μg of total RNA in a 20 μL reaction mixture, containing 2 μL of dNTP Mix (10 mM each), 0,5 μL of RNase Inhibitor, 40 U/μL, 0.5 μL of Oligo (dT) 12–18 (10 μL), 4 μL of 5× Reverse Transcriptase Buffer, 1 μL of RNA Template, 1 μL of RevertUPTM II Reverse Transcriptase and PCR Grade Water up to 20 μL. PCR reactions were performed with 50 ng of cDNA and GRS Taq DNA polymerase (Grisp, Porto, Portugal). The PCR program was the following: 95 °C for 5 min and 40 × (95 °C for 30 s, Ta for 30 s, 72 °C for 30 s), final elongation at 72 °C for 5 min.

### 2.4. qRT-PCR Analysis

To evaluate the expression of mitochondrial *sod2* and *gpx4* mRNAs, real-time qRT-PCR analysis was performed on the liver (5 specimens per site were singularly analysed). cDNAs for both target genes were amplified with the specific primers reported in Appendix A. To control for variation in the efficiency of cDNA synthesis and PCR amplification reactions, the housekeeping gene *gapdh* was used as a control and amplified with species-specific primers (Appendix A). qRT-PCR amplifications were carried out using the qPCRBIO SyGreen Mix Separate-ROX kit (PCR Biosystems, Wayne, PA, USA) and the following program: 95 °C for 2 min, 40 × (95 °C for 20 s, and 60 °C for 60 s), and then the dissociation stage 95 °C for 15 s, 60 °C for 1 min, 95 °C for 15 s, and 60 °C for 15 s.

### 2.5. Lipid Peroxidation (MDA) Assay

The lipid peroxidation assay was performed using the Abcam^®^ ab118970 Lipid Peroxidation (MDA) Assay Kit (Colorimetric/Fluorometric) (Waltham, MA, USA). This kit provides a convenient tool for the sensitive detection of malondialdehyde (MDA). In the lipid peroxidation assay, the MDA in the samples reacts with thiobarbituric acid (TBA) to generate an MDA–TBA adduct. The MDA–TBA adduct can be easily quantified colourimetrically (OD = 532 nm) or fluorometrically (Ex/Em = 532/553 nm). Each sample was at first weighed (the weight was kept between a range of 5–10 mg) and then washed with cold phosphate-buffered saline (PBS). Afterward, 150 μL of ultrapure water and 3 μL of BHT were added to each sample. Tissue samples were then vortexed, homogenised with pestles sitting on ice, and briefly centrifuged, after adding 163 μL of 2N perchloric acid. At the end of this process, 110 μL of supernatant was collected, and then 330 μL of TBA was added to both samples and standards. After incubation at 95 °C for 60 min, the samples were kept on ice for 10 min. The samples’ optical density (OD) was finally read on a microplate reader TECAN Infinite^®^ F200PRO (Männedorf, Switzerland) at 532 nm. Further calculations were performed following the manufacturer’s protocol. Data were normalised against total protein concentration. This assay was made only in *S. cephalus* liver (8 specimens per site were singularly analysed), *due to insufficient tissue availability for P. bonelli*.

### 2.6. Total Antioxidant Capacity (TAC) Assay

The total antioxidant capacity (TAC) assay was performed in *S. cephalus* and *P. bonelli* liver (4 and 3 specimens per site, respectively, were singularly analysed) using Abcam^®^ TAC 65329 Kit (Waltham, MA, USA). The kit for the TAC assay can measure the combination of both small antioxidant molecules and antioxidant enzymes, *also being able to discriminate the former with the use of* the Protein Mask present in the kit. For the analysis, 9 liver tissues of *P. bonelli* (3 specimens for each site) and 12 liver tissues of *S. cephalus* (4 specimens for each site) were analysed. To perform the assay, 10–15 mg of the liver was washed in cold PBS and then resuspended in 1 mL of cold PBS. The samples were homogenised using pestles sitting on ice and then the homogenates were incubated for 10 min on ice. The samples were finally centrifuged for 5 min at 4 °C at top speed to remove insoluble materials. After this step, the supernatant was collected and transferred to new tubes kept on ice. For TAC analysis, 100 μL of supernatant were mixed with 100 μL of Cu^2+^ working solution. For small molecules analysis, 100 μL of supernatant (previously diluted 1:1 with protein mask) were mixed with 100 μL of Cu^2+^ working solution. The protein mask prevents the reduction of Cu^2+^ by proteins, thus allowing the analysis of only small antioxidant molecules. The reduced Cu^+^ ion is chelated with a colorimetric probe giving an absorbance peak around 570 nm, proportional to the total antioxidant capacity. The samples’ OD was read on a microplate reader TECAN In-finite^®^ F200PRO (Männedorf, Switzerland). The enzymatic antioxidant capacity was calculated by the difference between TAC and small molecule antioxidant capacity. Data were normalised against total protein concentration.

### 2.7. Estimation of the Total Amount of Proteins

The total amount of proteins (TP) in the cellular extracts was assessed via the Folin phenol reagent method [22], using growing concentrations of bovine serum albumin as standard.

### 2.8. Hepatosomatic Index (HSI)

Total body weights (g) and lengths (cm) of all specimens were collected, together with the weight of each liver. From these measurements, the hepatosomatic index (HSI = weight of liver × 100/total fish weight) was determined. The analysed *P. bonelli* specimens were 9 from Site 1, and 10 from each of Sites 2 and 3. The analysed *S. cephalus* specimens were 8 from Site 1, 9 from Site 2, and 6 from Site 3.

### 2.9. Histology 

In total, 6 samples per species and site of the 10 originally sampled were evaluated according to morphological shifts in the liver tissue of *P. bonelli* as in *S. cephalus*. Liver samples were fixed with 4% paraformaldehyde in 0.1 M phosphate buffer (PB) until they were dehydrated in alcohol. The paraffin-embedded serial sections (5 µm) were cut with a Leica CM1850 microtome (Leica Biosystems, Deer Park, IL, USA) and stained with haematoxylin and eosin (H&E staining). The grade of fatty degeneration in the liver tissue was determined using a ranking system to describe the status of degeneration (severe, moderate, low, no degradation) which was defined before observation. Samples were analysed randomly without looking at the labelling to exclude an observation bias.

### 2.10. Statistical Analysis 

All data were expressed as the average of the analysed samples ± standard deviation (SD). Statistical analyses were performed with the PRIMER statistical program. One-way ANOVA was followed by the Student–Newman–Keuls test to assess significant differences (*p* < 0.05). The semi-quantitative comparison of the different histological sections was performed using Spearman rank correlation to describe the monotonic relation between pollution and grade of steatosis.

## 3. Results

### 3.1. qRT-PCR

The mRNA expression levels of *sod2* and *gpx4* in liver tissues have been analysed in both species, and the results are reported in Figure 1.

In *S. cephalus*, *sod2* was more expressed than *gpx4* (Figure 1A). The differences were between 600 and 1400 times (*p* < 0.001). The expression of this gene was statistically higher in the fish sampled from the low- and highly polluted sites with respect to those inhabiting the very low-polluted river (*p* < 0.05).

A diametrically opposite situation was found in *P. bonelli*, where *gpx4* was more expressed than *sod2* (Figure 1B). The differences were between 10 and 40 times (*p* < 0.001). 

Similar to the previous situation, the expression of this gene was statistically higher in the fish sampled from the low and highly polluted sites with respect to those living in the very low-polluted river (*p* < 0.05).

### 3.2. Lipid Peroxidation (MDA) Assay

The lipid peroxidation levels in the liver of fish sampled in both low and highly polluted sites were significantly lower (*p* < 0.05) than those measured in fish from the least polluted site, although the difference was only about 20% (18% for the low polluted site, 21% for the highly polluted one) (Figure 2).

### 3.3. Total Antioxidant Capacity (TAC) Assay

Figure 3A reports the results related to the total antioxidant capacity in the liver of *S. cephalus*. It is possible to notice that there were no statistically significant differences among the specimens sampled in the three sites.

Even in the liver of *P. bonelli*, the antioxidant capacity did not vary with respect to the different degrees of pollution by PFAS, despite the levels being slightly lower (but not statistically significant) in the fish sampled in the highly polluted site (Figure 3B).

### 3.4. Hepatosomatic Index (HSI)

Figure 4 shows the hepatosomatic index (HSI) values calculated on *P. bonelli* and *S. cephalus.* In *P. bonelli*; the only statistically significant difference was related to the specimens sampled in the low-polluted site, with values that were less than half of those found in the other analysed fish (*p* < 0.05). Concerning the HSI calculated on *S. cephalus*, there were statistically significant differences for all three experimental groups, with the lowest values in the fish sampled at the very low-polluted site and the highest in the specimens from the highly polluted site (*p* < 0.05). In particular, the latter exhibited values that were at least three times higher than those of other sampled fish.

### 3.5. Histopathological Diagnosis of the Liver 

*P. bonelli* showed a proportional increase in hepatocyte cytoplasmic vacuolisation with the increase in PFOS concentration found in the organism (4.64 µg/kg in the very low-polluted, 10.78 µg/kg in the low-polluted, and 112.41 µg/kg in the highly polluted site) (Figure 5). Two out of four analysed fish from the low-polluted site showed severe microvesicular steatosis. In contrast, all five analysed samples from the highly polluted site showed severe macrovesicular steatosis, characterised by single fat filed vacuole. The level of steatosis highly correlated with the level of PFOS accumulation in *P. bonelli* (ρ = 0.721), especially in specimens from the highly polluted area, which showed high-level macrovesicular steatosis in the liver. In *S. cephalus*, there was very limited steatosis unrelated to the accumulated amount of PFOS (ρ = −0.315) (Figure 5).

## 4. Discussion

In recent decades, the Veneto provinces of Vicenza, Verona, and Padova have been affected by one of the most extensive PFAS contamination in the world [10]. Most of the research efforts on the physiological effects of PFAS have been focused on human health. Many studies have shown that the exposure of the human population to these pollutants could lead to several pathologies. Since PFAS can interact with many molecular components of the cell, they can also interfere with different metabolic pathways [23,24], leading to an increase in ROS production and pathologies such as immunotoxicity, neurotoxicity, and alterations of the reproductive system [25]. It has been shown that PFAS can also cause endocrine disruption and alteration of lipid metabolism [3,26,27]. 

By contrast, very little is known about the physiological effects that PFAS could produce on non-human organisms. Even less is known thus far about the interactive effects of different compounds of PFAS when present simultaneously [28]. This study aimed at verifying the physiological effects and responses induced by PFAS in freshwater fish.

The approach to this study was truly multidisciplinary, integrating biochemistry, molecular biology, and histology at locations of a fluvial system with mapped pollution levels. The expression of two genes, *gpx4* and *sod2*, were analysed in fish sampled in three different aquatic sites to evaluate the antioxidant response in mitochondria, one of the main endogenous sources of ROS and the targets of PFAS toxicity, where the encoded enzymes are specifically expressed. Two biochemical assays were used to evaluate lipid peroxidation and total antioxidant capacity. The liver was chosen as the primary site of PFAS accumulation.

From the gene expression analysis, it is possible to verify the activation of the antioxidant gene expressions in the liver that is related to the exposure to PFAS and the accumulation of these pollutants inside the tissues. In *S. cephalus,* the PFAS induced an increase in *sod2* expression. In particular, going from a very low level of environmental pollution to a slightly higher level, the mRNA expression for this gene doubled. This higher level of expression remained almost unchanged, even in the specimens sampled in the highly polluted site by PFAS.

These results are compatible with an increase in superoxide radical formation at the mitochondrial level since superoxide dismutase is the specific scavenger of this ROS and the very specific cellular localisation of SOD2 [29]. However, they also indicate that liver cells do not seem to respond proportionally to the concentration of PFAS, although it should promote a greater risk of oxidative stress. It is possible to hypothesise that the measured expression levels of SOD2 are already protective against this risk. In fact, these animals have been chronically exposed to PFAS and, therefore, had time to balance the antioxidant response through negative feedback control, which is characteristic of the expression of proteins that were part of the antioxidant system [30]. To verify these hypotheses, it should be useful to evaluate the effect of acute stress, for example, through in vivo experiments under controlled laboratory conditions.

The very limited expression of GPX4 seems inexplicable, given that this enzyme should be involved in the elimination of hydrogen peroxide produced by SOD2. A hypothesis is that other components of the antioxidant defencee system may be activated in specimens of *S. cephalus* after PFAS exposure, such as *Prdx3* and *Prdx5*, two isoforms of peroxiredoxins specifically expressed in mitochondria. Peroxiredoxins are enzymes that act in a specific way by reducing hydrogen peroxide and other peroxides. The upregulation of mitochondrial peroxiredoxins could compensate for the limited expression of GPX4. In order to verify this hypothesis, it will be essential to evaluate the gene expression of these antioxidant enzymes in the liver.

A diametrically opposite situation was found in *P. bonelli*, where the presence of PFAS pollution, induced an increase in *gpx4* expression. The mRNA levels of this gene tripled as PFAS concentration increased going from a very low level of pollution to a slightly higher level. This higher level of expression remained almost unchanged even in the specimens sampled in the highly polluted site. The obtained results are in line with a possible increase in the formation of peroxides at the mitochondrial level. In fact, GPX4 is specifically expressed in mitochondria and provides a specific scavenging action against this type of ROS [31]. It remains to clarify whether the formation of hydrogen peroxide occurs by dielectronic reduction of molecular oxygen or by monoelectronic reduction of the superoxide radical. However, limited levels of *sod2* induction seem to exclude the second hypothesis.

As a response to the greater environmental presence of PFAS and higher hepatic accumulation of these substances, liver cells do not seem more responsive. Similar to what has been proposed for *S. cephalus*, we can hypothesise that the expression levels of *gpx4* are already protected against the risk of oxidative stress. However, also in specimens of *P. bonelli* living in the very high polluted river, the upregulation of mitochondrial peroxiredoxins could explain the limited expression of GPX4.

As for the biochemical analyses, the data on lipid peroxidation do not correlate with the levels of PFAS found in the three rivers. In particular, a supposed increase in ROS formation should be associated with an increase in the accumulation of PFAS, and therefore, an increase in lipid peroxidation could have been expected. Instead, our results indicated a little decrease in this parameter as the level of environmental pollution increased. This evidence can be explained by the increase in the expression of antioxidant enzymes previously discussed and the different timing of exposure. During the acute phase of exposure to PFAS, there could have been a likely increase in ROS formation. This may have led to an increase in lipid peroxidation and, consequently, to the activation of genes coding antioxidant proteins, especially at the level of mitochondria, to counteract the damage to cell membranes. However, this study shows that, in the chronic exposure of fish from the contaminated sites, the maintenance of high levels of the antioxidant response may favour a prompt elimination of ROS produced by PFAS, also leading to a lower-than-normal level of lipid peroxidation.

Regarding the liver TAC, which is the result of post-transcriptional expression of antioxidant proteins and molecules, the absence of statistically significant differences among the experimental groups could be mainly due to a lack of activation of the antioxidant defences at the cytoplasmic level. The target of PFAS toxicity is mitochondria, and the expression of antioxidant enzymes is proven. However, the amount of antioxidant protein biosynthesised by these organelles is lower than the one produced at the cytoplasmic level, where many more antioxidant enzymes and isoforms are expressed, possibly covering the increase in antioxidant activity in the mitochondria.

The histological analyses show a differential response of the two species towards PFAS exposure. Specifically, it is possible to notice a positive correlation between lipid vacuolisation (steatosis) in *P. bonelli* but not in *S. cephalus*. 

A positive correlation between hepatocyte vacuolisation and PFAS accumulation in the liver was shown also in eels exposed to PFOA [32], zebrafish exposed to PFOS [31], and rats exposed to PFOA and PFOS [33]. The increase in vacuolation could be considered a defence mechanism against the toxic effects of PFAS. In fact, it is known that these compounds show their toxicity against cytoplasmic proteins, which are purely hydrophilic molecules. By increasing the amount of fat inside the cell, the hydrophilicity of the intracellular environment is reduced, making the interaction between PFAS and the protein component less probable. The accumulation of lipids in *P. bonelli* could lead to a reduction in the specific weight of the tissue and, therefore, to a reduction in HSI, as actually occurred in the specimens of this species exposed to the low-polluted environment. However, it seems that this defence mechanism is insufficient when there is an increase in the accumulation of PFAS in the organ, as it occurred in specimens exposed to a highly polluted environment (Schumann et al., “Biomonitoring the effect of accumulating perfluorochemicals on physiological responses in freshwater fish from different ecological niches”, submitted). In this case, they also require a proliferation of hepatocytes to implement the detoxifying action, which brings the HSI back to increase.

On the other hand, steatosis does not seem to be a defence mechanism that characterises *S. cephalus*, probably because this species is able to accumulate much more PFAS in its own tissues and, in particular, in the liver (Schumann et al., “Biomonitoring the effect of accumulating perfluorochemicals on physiological responses in freshwater fish from different ecological niches”, submitted). The need for liver mass increase becomes more dominant, with a consequent increase in HSI, as verified in specimens from low- and especially high-polluted sites. Hepatic hyperplasia and hypertrophy, together with steatosis, are a consequence of the accumulation of PFAS in this organ well-known in mammals [34,35] but also found in zebrafish chronically exposed to PFAS [36,37].

This study also shows that vacuolisation is species-dependent and, in the case of PFAS pollution, might correlate with the habits of the species. Decreased vacuolisation can result from a reduction in the cytoplasmic glycogen and/or lipids caused by insufficient energy intake or the use of resources due to glucocorticoid-induced glycogenolysis [33]. It is, therefore, presumable to hypothesise that this species, notoriously potamodromous, responded to xenobiotic stress through greater use of energy resources. On the contrary, in *P. bonelli*, a sedentary species, the PFAS stress seems to have induced the phenomenon of steatosis. This phenomenon is commonly associated with overnutrition or chronic toxicity. It has already been observed in other studies, for example, in zebrafish fry chronically exposed to PFOS [33,38].

## 5. Conclusions

The results suggest that PFAS chronic exposure induces some physiological responses in the target species, at both cellular and tissue scales. The risk of oxidative stress seems to be kept under control by the antioxidant system by means of gene activation at the mitochondrial level. Furthermore, the histological analysis suggests an additional interesting protective mechanism against damage to the protein component based on lipid vacuolisation.

This study provided new data on the effects of chronic exposure to PFAS on two freshwater fish species. The multidisciplinary approach led to interesting results on the different sensitivity of the two species and on the different physiological responses that they can elicit when their surrounding environment changes. Results also highlighted that stress responses are very species-dependent, and this is something that will certainly prompt future studies.

The results obtained in this study could also provide useful information on the impact of PFAS within the trophic chains and consequently within products intended for food consumption. This issue is now more relevant than ever since, on 20 September 2021, after a legal battle with the Veneto Region, the results of food analysis sampled from the so-called “red zone” were published. These analyses show that PFAS contamination concerns drinking water as well as foods of various kinds. This is of particular concern since recent research has identified the consumption of food, particularly fish and shellfish, as one of the main roads of exposure to PFAS [39].

From the present research, it also emerges that field studies are more complex to interpret since they do not allow definite conclusions about the causality of the observed effects. For example, it must be taken into account that per- and poly-fluoroalkyl substances may not be the only pollutants present in the three sampled sites, although this is an assumption on which the present study is based. In this respect, the obtained data help to formulate hypotheses and to indicate approaches that will likely promote further investigation in the near future. In light of the growing knowledge about this topic, new urgent actions appear unavoidable to reduce the potential damage [40]. It is now apparent that PFAS pollution routes fast so that it has reached the Venice lagoon already, thus affecting the lagoon organisms and consequently the regional economy.

Most of the studies that focus on PFAS-induced effects in fish always use very high concentrations of single PFAS compounds. However, as the present study highlights possible toxic responses at low PFAS concentrations, it appears extremely interesting to address future studies using similar concentrations. 

New laboratory experiments, carried out under controlled conditions, will certainly be necessary to establish the correlation between physiological changes, histological damages, and exposure to a specific pollutant or class of pollutants. Even if studies on the toxicity of single compounds are certainly necessary to gain basic toxicological information, the real exposures to which humans and other animals are subjected are characterised by a coexistence of different chemical compounds. For this reason, focusing on the toxicity of mixed chemicals could be a promising approach to this topic that may likely improve future health and environmental risk assessment [41].

## Figures and Tables

**Figure 1 antioxidants-11-01115-f001:**
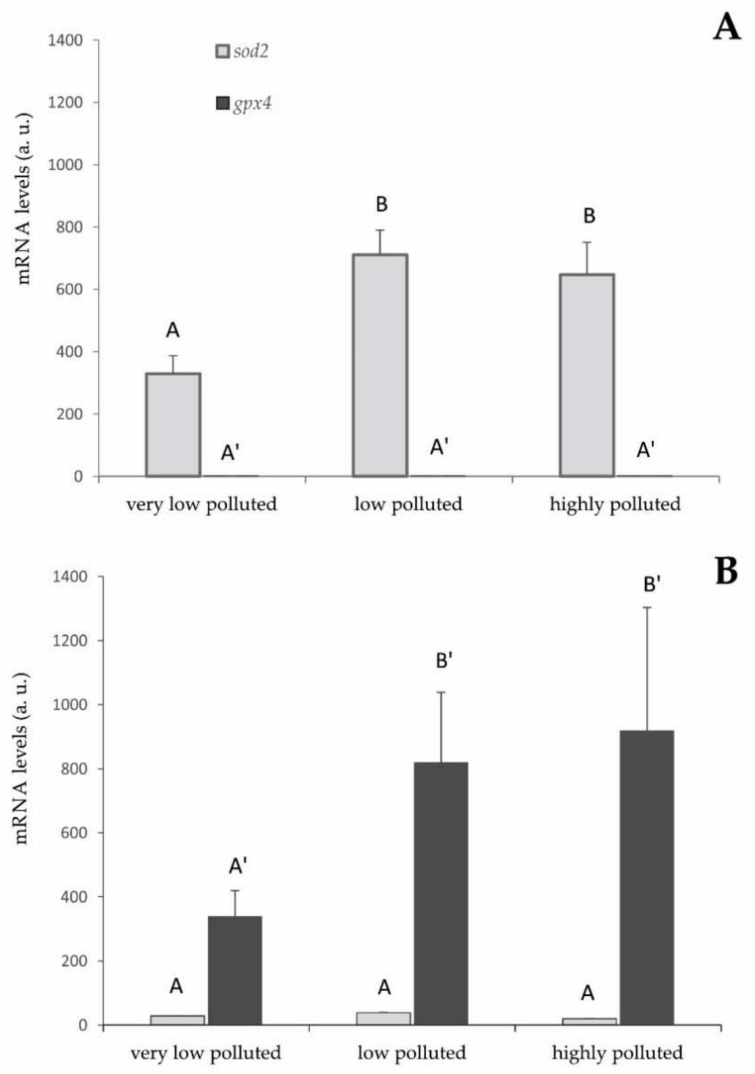
gpx4 and sod2 mRNA levels in (**A**) S. cephalus and (**B**) P. bonelli liver. Values (arbitrary units) are indicated as mean ± SD. Transcription levels were normalised to the gapdh housekeeping gene. Different letters correspond to significant statistical differences (*p* < 0.05) among different sites (Student–Newman–Keuls test). Five specimens per site were singularly analysed.

**Figure 2 antioxidants-11-01115-f002:**
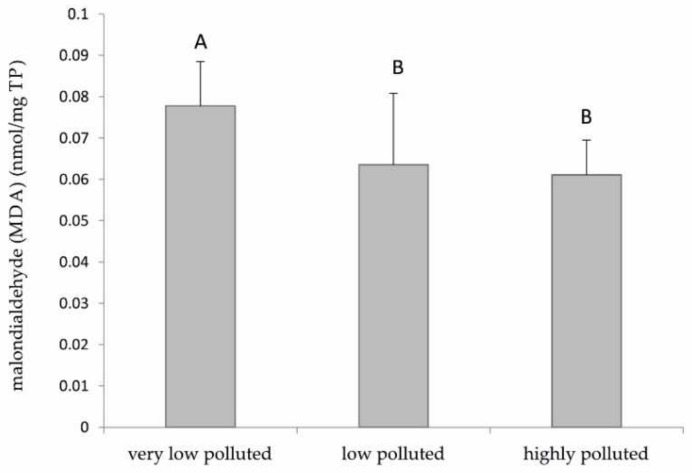
Malondialdehyde mean concentrations in *S. cephalus* liver. Values (arbitrary units) are indicated as mean ± SD. Different letters correspond to significant statistical differences (*p* < 0.05) among different sites (Student–Newman–Keuls test). Eight specimens per site were singularly analysed.

**Figure 3 antioxidants-11-01115-f003:**
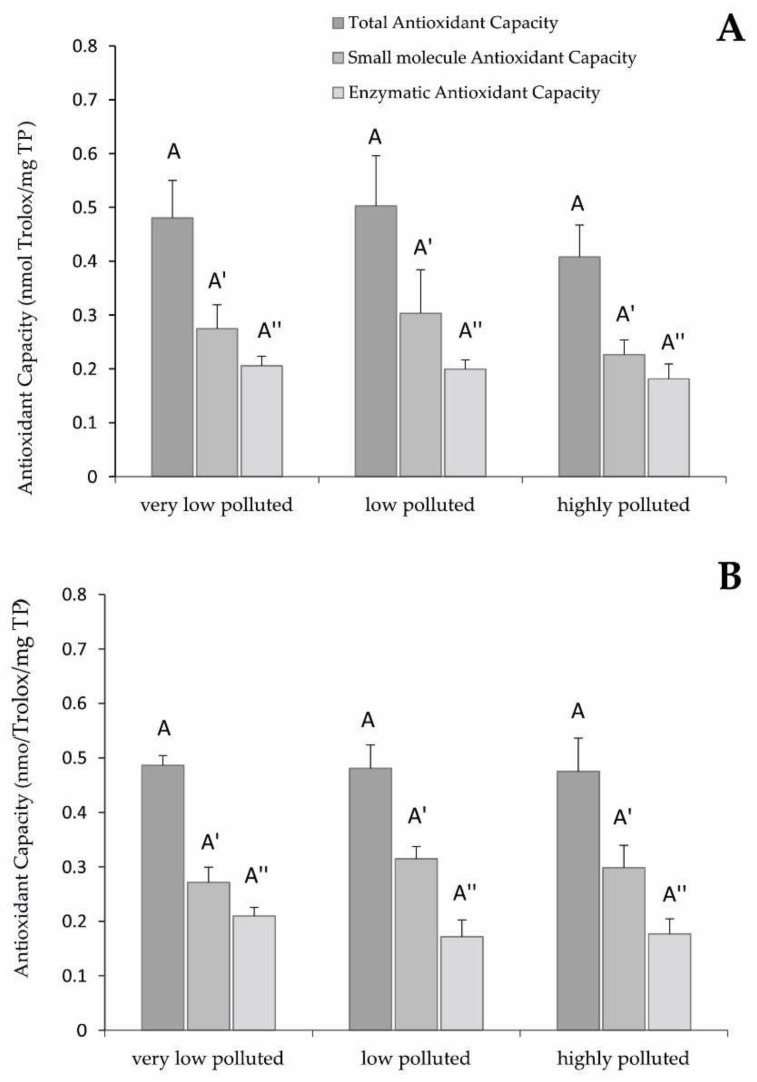
Total antioxidant capacity (sum of small molecules antioxidant capacity and enzymatic antioxidant capacity) of (**A**) *S. cephalus* and (**B**) *P. bonelli* liver. Values (nanomoles of Trolox per milligrams of total proteins) are indicated as mean ± SD. Different letters correspond to significant statistical differences (*p* < 0.005) among different sites (ANOVA test). *P. bonelli*: three specimens per site. *S. cephalus*: four specimens per site were singularly analysed.

**Figure 4 antioxidants-11-01115-f004:**
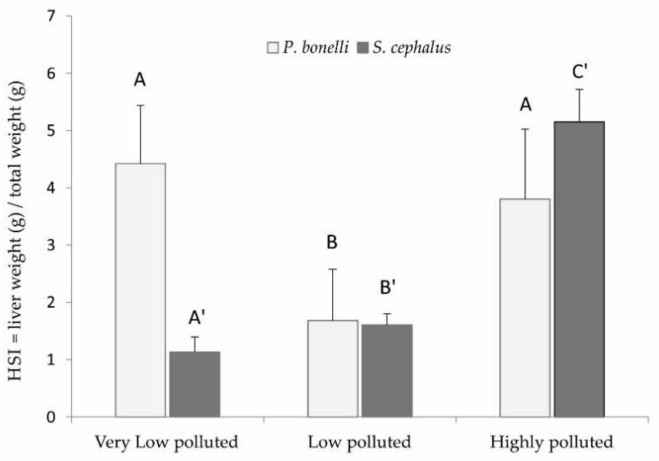
Hepatosomatic index in *P. bonelli* and *S. cephalus*. Values (grams of liver weight per grams of total body weight) are indicated as mean ± SD. The three means were statistically compared with each other. Different letters correspond to significant statistical differences (*p* < 0.05) among different sites (Student–Newman–Keuls test). *P. bonelli* specimens: Site 1 n = 9, Sites 2 and 3 n = 10. *S. cephalus* specimens: Site 1 n = 8, Site 2 n = 9, and Site 3 n = 6.

**Figure 5 antioxidants-11-01115-f005:**
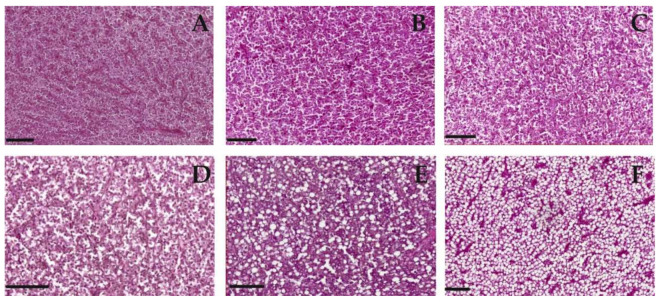
Lipid vacuolisation in liver of (**A**–**C**) *S. cephalus* and (**D**,**E**) *P. bonelli* caught in the three different rivers with (**A**,**D**) very low, (**B**,**E**) low, and (**C**,**F**) high concentration of PFAS. Increasing vacuolisation was observed only in *P. bonelli* liver, and it was proportional to increasing hepatic PFAS concentration (from 139.25 µg/kg per whole body weight). Scale bars indicate 50 µm. Six specimens per site.

## Data Availability

The data are contained within the article and Appendix A.

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
