# Peer review of "Antioxidant Responses Induced by PFAS Exposure in Freshwater Fish in the Veneto Region"

_antioxidants, 2022, doi:10.3390/antiox11061115_

Round 1
Reviewer 1 Report
Dear Editor, in the submitted work the antioxidant responses induced by pfas exposure in fresh water fish in the Veneto region. The paper is well organized and contains some new and import results. For this reason, I propose to be accepted for publication. In following you can find some specific comments.
There are many typographical mistakes and should be corrected.
Which is the origin of PFAS pollution? Are close to these regions some companies?
Any estimation of this pollution to human health.
Author Response
Dear Editor, in the submitted work the antioxidant responses induced by pfas exposure in fresh water fish in the Veneto region. The paper is well organized and contains some new and import results. For this reason, I propose to be accepted for publication. In following you can find some specific comments.
There are many typographical mistakes and should be corrected.
The manuscript has been revised to resolve some typos.
Which is the origin of PFAS pollution? Are close to these regions some companies?
In 2013 the Regional Agency for Environmental Prevention and Protection of Veneto (ARPAV) identified the Miteni factory, one of the major producers of per- and poly- fluoroalkyl substances since 1970s, as the main responsible of the contamination. This factory is located in Trissino (VI), in the same province of the selected sampling sites.
Any estimation of this pollution to human health.
The median value of the serum concentration of PFAS of people exposed to PFAS was 8 times higher than the median value of those not exposed.
The Regional Administration has prepared an ongoing health surveillance plan for people residing in the Area Impacted by PFAS (30 Municipalities) in order to favour the prevention, early diagnosis and treatment of some chronic diseases with epidemiological evidence of association with exposures to PFAS, i.e. dyslipidemia, hypertension, diabetes mellitus, liver dysfunction, metabolic syndrome, renal dysfunction and thyroid disorders.
If required, we can add this information in the Introduction.
Reviewer 2 Report
The MS of Piva et al., “Antioxidant Responses Induced By PFAS Exposure In Freshwater Fish In The Veneto Region” is in general well written and structured. Some parts could be more detailed and improved. The topic fits with the scopes of Antioxidant journal and can be of interest for the readership.
Following, some comments to improve the manuscript:
_ L 83-90: I suggest moving this paragraph in Discussion section.
_ L 91-96: I suggest moving this paragraph in Conclusion section.
_ Materials and methods:
1)why Authors did not perform a monthly or seasonal monitoring of the three sites investigated? 2)Why Authors did not assess the catalase expression? Catalase is a pivotal enzyme counteracting hydrogen peroxide and following oxidative effects. I suggest considering assessing this enzyme expression to further improve the scientific value of the study. 3)Why Authors performed MDA assay only in S. cephalus? This should be clearly specified in the paper.
My apologies, I did not understand how many fish were sampled and then used for analysis, since results for each biochemical or histological analyzes reported different fish numbers. I suggest clearly specify how many fish were sampled and used for each analysis, as well as report if fish were pooled or singularly analyzed.
Overall, I recommend publication after major revisions.
Author Response
The MS of Piva et al., “Antioxidant Responses Induced By PFAS Exposure In Freshwater Fish In The Veneto Region” is in general well written and structured. Some parts could be more detailed and improved. The topic fits with the scopes of Antioxidant journal and can be of interest for the readership.
Following, some comments to improve the manuscript:
_ L 83-90: I suggest moving this paragraph in Discussion section.
This paragraph has been moved in Discussion section
_ L 91-96: I suggest moving this paragraph in Conclusion section.
This paragraph has been moved in Conclusion section
_ Materials and methods:
1)why Authors did not perform a monthly or seasonal monitoring of the three sites investigated?
The aim of the study was to study the physiological responses in fish chronically exposed to PFAS. A biomonitoring activity should be interesting, especially in relation to highlight variations in PFAS tissue concentrations. Unfortunately, correlating any variations of this chemical parameter with the stress condition would be very difficult, as in the different seasons the effect of the PFAS would be added to that of other environmental variations such as those of temperature and oxygenation of the water, correlated to the different flow rate of waterways. Furthermore, physiological conditions such as the reproductive phase could also cause further variations in biochemical and biomolecular parameters that are difficult to discriminate.
2)Why Authors did not assess the catalase expression? Catalase is a pivotal enzyme counteracting hydrogen peroxide and following oxidative effects. I suggest considering assessing this enzyme expression to further improve the scientific value of the study.
Many other antioxidant enzymes could be considered, such as the 6 isoforms of peroxiredoxins, the other 2 isoforms of superoxide dismutase and the other 7 isoforms of glutathione peroxidase. The choice to limit the study to SOD2 and GPX4 is due to the fact that these two enzymes act to protect the mitochondria, the cellular structure that is universally recognized as the main target of PFAS toxicity, from oxidative stress. Catalase is a peroxisomal enzyme, therefore it is not primarily involved in the detoxification of ROS produced by PFAS. We are certainly interested in evaluating also the other antioxidant enzymes in the future, to achieve a complete picture of the activation of the antioxidant defense cellular system by PFAS, not only with regard to the expression of mRNA, but also to that of the active protein. However, in order to do this it would be necessary to sacrifice a much greater number of specimens sampled in the wild, which is hardly compatible with environmental protection standards.
3)Why Authors performed MDA assay only in S. cephalus? This should be clearly specified in the paper.
This test was performed only in the liver of S. cephalus (8 specimens per site), due to insufficient tissue availability for P. bonelli. This information has been added on page 4, lines 173-174.
My apologies, I did not understand how many fish were sampled and then used for analysis, since results for each biochemical or histological analyzes reported different fish numbers. I suggest clearly specify how many fish were sampled and used for each analysis, as well as report if fish were pooled or singularly analyzed.
In the Materials and Methods section (page 3, lines 108-109) it was indicated that “Ten specimens of each species, S. cephalus, and P. bonelli, were sampled from each of the three sites.”. For each analysis, we report the number of specimens analysed. We specify them better now and we also indicate that fish were singularly analyzed.
Overall, I recommend publication after major revisions.
Round 2
Reviewer 2 Report
The MS was revised according suggestions of reviewer. The improved MS is thus suitable for publication in journal Antioxidants.